# A teacher-student framework
# to distill future trajectories

**Alexander Neitz[1,*]   Giambattista Parascandolo[1,2,*]   Bernhard Schölkopf[1,2]**
[1]MPI for Intelligent Systems, Tübingen,   [2]ETH, Zürich,
*equal contribution

## Abstract

By learning to predict trajectories of dynamical systems, model-based methods can make extensive use of all observations from past experience. However, due to partial observability, stochasticity, compounding errors, and irrelevant dynamics, training to predict observations explicitly often results in poor models. Model-free techniques try to side-step the problem by learning to predict values directly. While breaking the explicit dependency on future observations can result in strong performance, this usually comes at the cost of low sample efficiency, as the abundant information about the dynamics contained in future observations goes unused. Here we take a step back from both approaches: Instead of hand-designing how trajectories should be incorporated, a *teacher* network learns to extract relevant information from the trajectories and to distill it into target activations which guide a *student* model that can only observe the present. The teacher is trained with meta-gradients to maximize the student's performance on a validation set. Our approach performs well on tasks that are difficult for model-free and model-based methods, and we study the role of every component through ablation studies.

## 1 Introduction

The ability to learn models of the world has long been argued to be an important ability of intelligent agents. An open and actively researched question is how to learn world models at the right level of abstraction. This paper argues, as others have before, that model-based and model-free methods lie on a spectrum in which advantages and disadvantages of either approach can be traded off against each other, and that there is an optimal compromise for every task. Predicting future observations allows extensive use of all observations from previous experiences during training, and to swiftly transfer to a new reward if the learned model is accurate. However, due to partial observability, stochasticity, irrelevant dynamics and compounding errors in planning, model-based methods tend to be outperformed asymptotically (Pong et al., 2018; Chua et al., 2018). On the other end of the spectrum, purely model-free methods use the scalar reward as the only source of learning signal. By avoiding the potentially impossible task of explicitly modeling the environment, model-free methods can often achieve substantially better performance in complex environments (Vinyals et al., 2019; OpenAI et al., 2019). However, this comes at the cost of extreme sample inefficiency, as only predicting rewards throws away useful information contained in the sequences of future observations.

What is the right way to incorporate information from trajectories that are associated with the inputs? In this paper we take a step back: Instead of trying to answer this question ourselves by hand-designing what information should be taken into consideration and how, we let a model *learn* how to make use of the data. Depending on what works well within the setting, the model should learn *if and how* to learn from the trajectories available at training time. We will adopt a teacher-student setting: a *teacher* network learns to extract relevant information from the trajectories, and distills it into target activations to guide a *student* network.[1] A sketch of our approach can be found in Figure 1, next to prototypical computational graphs used to integrate trajectory information in most model-free and model-based methods. Future trajectories can be seen as being a form of *privileged information* Vapnik and Vashist (2009), i.e. data available at training time which provides additional information but is not available at test time.

---

[1]Note that the term *distillation* is often used in the context of "distilling a large model into a smaller one" (Hinton et al., 2015), but in this context we talk about distilling a trajectory into vectors used as target activations.

**Contributions** The main contribution of this paper is the proposal of a generic method to extract relevant signal from privileged information, specifically trajectories of future observations. We present an instantiation of this approach called Learning to Distill Trajectories (LDT) and an empirical analysis of it.

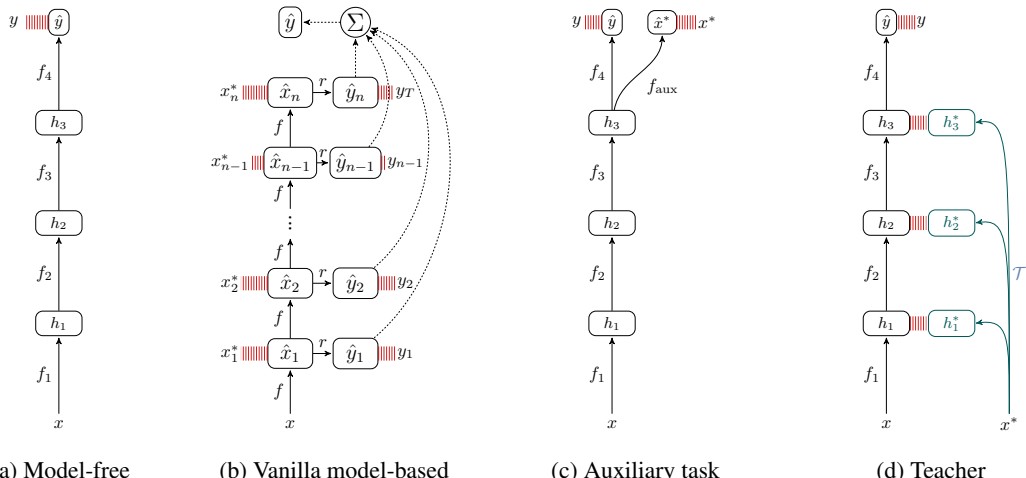

| (a) Model-free | (b) Vanilla model-based | (c) Auxiliary task | (d) Teacher |

Figure 1: Comparison of architectures. The data generator is a Markov reward process (no actions) with an episode length of $n$. $x$ denotes the initial observation. $y = \sum_i y_i$ is the $n$-step return (no bootstrapping). $x^* = (x_1^*, x_2^*, ..., x_n^*)$ is the trajectory of observations (privileged data). Model activations and predictions are displayed boxed. Losses are displayed as red lines. Solid edges denote learned functions. Dotted edges denote fixed functions.

## 2 RELATED WORK

Efficiently making use of signal from trajectories is an actively researched topic. The technique of bootstrapping in TD-learning (Sutton, 1988) uses future observations to reduce the variance of value function approximations. However, in its basic form, bootstrapping provides learning signal only through a scalar bottleneck, potentially missing out on rich additional sources of learning signal. Another approach to extract additional training signal from observations is the framework of Generalized Value Functions (Sutton et al., 2011), which has been argued to be able to bridge the gap between model-free and model-based methods as well. A similar interpretation can be given to the technique of successor representations (Dayan, 1993).

A number of methods have been proposed that try to leverage the strengths of both model-free and model-based methods, among them Racanière et al. (2017), who learn generative models of the environment and fuse predicted rollouts with a model-free network path. In a different line of research, Silver et al. (2017) and Oh et al. (2017) show that value prediction can be improved by incorporating dynamical structure and planning computation into the function approximators. Guez et al. (2019) investigate to what extent agents can learn implicit dynamics models which allow them to solve planning tasks effectively, using only model-free methods. Similarly to LDT, those models can learn their own utility-based state abstractions and can even be temporally abstract to some extent. One difference of these approaches to LDT is that they use reward as their only learning signal without making direct use of future observations when training the predictor.

The meta-gradient approach presented in this paper can be used more generally for problems in the framework of *learning using privileged information* (LUPI, (Vapnik and Vashist, 2009; Lopez-Paz et al., 2016)), where privileged information is additional context about the data that is available at training time but not at test time. Hindsight information such as the trajectories in a value-prediction task falls into this category.

There are a variety of representation learning approaches which can learn to extract learning signal from trajectories. Jaderberg et al. (2016) demonstrate that the performance of RL agents can be

improved significantly by training the agent on additional prediction and control tasks in addition to the original task. Du et al. (2018) use gradient similarity as a means to determine whether an auxiliary loss is helpful or detrimental for the downstream task. Oord et al. (2018) introduce a method based on contrastive learning. They, as well as multiple follow-up studies, show that the representations learned in this way are helpful for downstream tasks in a variety of settings.

Buesing et al. (2018) present ways to learn efficient dynamical models which do not need to predict future observations at inference time. Recently, Schrittwieser et al. (2019) introduced an RL agent that learns an abstract model of the environment and uses it to achieve strong performance on several challenging tasks. Similarly to our motivation, their model is not required to produce future observations. Meta-learning approaches have recently been shown to be successful as a technique to achieve fast task adaptation (Finn et al., 2017), strong unsupervised learning (Metz et al., 2019), and to improve RL (Xu et al., 2018). Similar to this paper in motivation is the recent work by Guez et al. (2020) which also investigates how privileged hindsight information can be leveraged for value estimation. The difference to LDT is how the trajectory information is incorporated. Their approach has the advantage of not needing second-order gradients. At the same time, LDT naturally avoids the problem of the label being easily predictable from the hindsight data — the teacher is trained to present it to the student in such a way that it empirically improves the student's performance on held-out data. Veeriah et al. (2019) use meta-gradients to derive useful auxiliary tasks in the form of generalized value functions. In contrast, we use a teacher network that learns to provide target activations for a student neural network based on privileged information.

## 3  META-LEARNING A DYNAMICS TEACHER

Here we describe our approach of jointly learning a teacher and a student.[2] While our approach applies to the generic setting of *learning using privileged information* (Vapnik and Vashist, 2009), here we will focus on the special case of a prediction task with an underlying dynamical system.

### 3.1  LEARNING TASK

We are considering learning problems in which we have to make a prediction about some property of the future state of a dynamical system, given observations up to the current state. Our method particularly applies to systems in which both the function that relates the current observation to the label as well as the function that predicts the next observation from the current one are hard to learn, making it difficult for both model-free and model-based methods respectively.

To make the explanation more concrete, we will use the practical problem of medical decision-making as a running example to which we can relate the definitions we used, similar to a motivating example from Vapnik and Vashist (2009): given the history of measurements (biopsies, blood-pressure, etc.) on a given patient and the treatment assignment, we want to predict whether the patient will recover or not.

The input $x \in \mathcal{X}$ of our learning task is some observation of the system state $s_t \in \mathbf{S}$ before and including time step[3] $t \in \mathbb{Z}$. In our running example, $s_t$ can be considered the detailed physical state of the patient, which is not directly observable. The observations $x$ include potentially multi-modal data such as x-ray images, vital sign measurements, oncologist reports, etc. The system is governed by an unknown dynamical law $f : \mathbf{S} \to \mathbf{S}$ — in our example, the dynamics are physical equations that determine the evolution of all cells in the body. The prediction target $y \in \mathcal{Y}$ is some function of a future state $s_T = f^{T-t}(s_t)$, separated from $t$ by $T - t$ time steps: $y = g(s_T)$. In our running example, a prediction target could be the binary indicator of whether the patient will recover within some time frame. Note that $T$ could vary from one example to the next. In addition to the initial observation, we have access to the trajectory $x^* = (x_\tau)_{\tau=t+1..T}$ at training (but not test) time. In our running example, the trajectory includes all measurements from the patient *after* the treatment decision has been made. This information is available in a dataset of past patients (in hindsight), but not in any novel situation.

---

[2]Note that unlike in some related work, the teacher in our task is *not* a copy of the student network, but can have a completely different architecture.

[3]For simplicity, our dynamical system is time-discrete, but this assumption is not important for what follows.

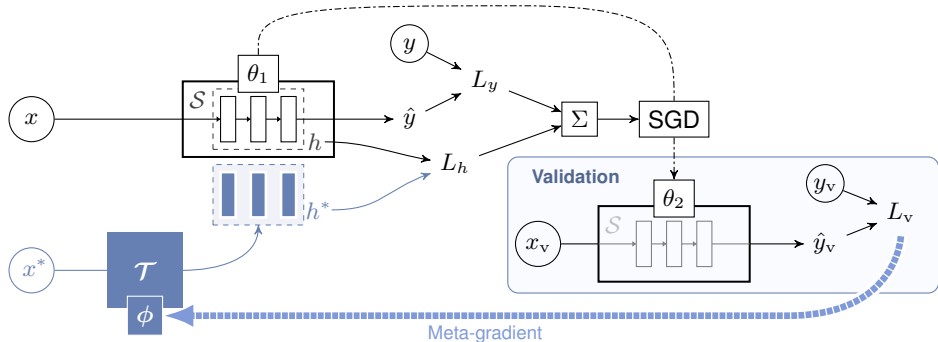

Figure 2: Visualization of the LDT framework for the special case of $n = 1$. Circled nodes are part of the dataset. $x$ denotes the input, $x^*$ is the privileged data, $y$ is the label. $\mathcal{S}$ is the student network with parameters $\theta$, $\mathcal{T}$ is the teacher network with parameters $\phi$.

### 3.2 SUPERVISION OF INTERNAL ACTIVATIONS

A straightforward approach to solve the learning task which takes into account the trajectory information, would be to train a state-space-model (SSM) $\hat{f}$, consisting of a dynamical model and a decoder. The SSM is trained to maximize the likelihood of the observed trajectories in the training set, conditioned on the observed initial observation. Ideally, the induced $\hat{f}$ closely resembles $f$, such that at test time, we can use it to generate an estimate of the rollout and infer the label from it. A potential drawback of this approach is that learning a full SSM could be more difficult than necessary. There may be many details of the dynamics that are both difficult to model and unimportant for the classification tasks. One example for this is the precise timing of events. As argued by Neitz et al. (2018); Jayaraman et al. (2019), there are situations in which it is easy to predict a sequence of events where each event follows a previous one, but hard to predict the exact timing of those events. Moreover, an SSM typically requires rendering observations at training time, which may be difficult to learn and computationally expensive to execute.

In the running example from Section 3.1, it seems challenging and wasteful to predict all future observations in detail, as it would require modeling a complicated distribution over data such as X-ray images or doctor reports written in natural language. Ideally, we would like a model to learn how to extract the relevant information from these data efficiently.

We propose to relax the requirement of fitting the dynamics precisely. The teacher can decide to omit properties of the observations that are not needed and omit time steps that can be skipped. It could also change the order of computation and let the student compute independently evolving sub-mechanisms sequentially, even if they evolved in parallel in the actual data generating process. In addition to potentially simplifying the learning problem, this could have the additional benefit of gaining computational efficiency. For example, modeling detailed pixel observations may be computationally wasteful, as argued by Buesing et al. (2018) and Oord et al. (2018).

### 3.3 STUDENT-TEACHER SETUP

We propose a student-teacher setup with two neural networks, as shown in Figure 2. The *student network* $\mathcal{S}$, parameterized by weights $\theta$, is the network that attempts to predict the quantity of interest $y$ (for instance a cumulative reward or value). Its input is $x \in \mathcal{X}$, and its output is $\mathcal{S}(x) = \hat{y} \in \mathcal{Y}$. In computing $\hat{y}$, it produces a sequence of internal activations $(h_1, ..., h_N)$, one for each of its $N$ hidden layers. Each $h_k$ is a vector whose size is the number of neurons of the corresponding hidden layer. The student's goal is to minimize the generalization loss $\mathbb{E}_{x,y \sim P_{\text{test}}}[\mathcal{L}_y(S(x), y)]$ for some loss function $\mathcal{L}_y : \mathcal{Y} \times \mathcal{Y} \to \mathbb{R}$.

The *teacher network* $\mathcal{T}$, parametrized by weights $\phi$, is only used at training time, not at test time. It reads the observations of the rollout $x^* = (x_\tau)_{\tau=t+1..T}$ corresponding to the current training example, and outputs supervision signals $(h_1^*, ..., h_N^*)$.

The target activations produced by the teacher's supervision result in another loss for the student, the *teaching loss*, defined as $L_h = \sum_k \mathcal{L}_h(h_k, h_k^*)$. $\mathcal{L}_h$ denotes the teaching loss function which, given

a pre-activation and a supervision signal, produces a scalar value. It can be chosen to be any common loss function. Note however that in general, $\mathcal{L}_h$ could combine its inputs in an arbitrary way, as long as it is differentiable and produces a scalar. In particular, $h_k$ and $h_k^*$ are not required to have the same dimensionality. For example, in our specific instantiation described in Section 4, $h^*$ contains masking weights to gate the teaching signal. The total student training loss is $L_{\text{train}} = \alpha L_h + L_y$, where $L_y$ is the *label loss*, e.g. the cross-entropy error between predictions and true labels. $\alpha \in \mathbb{R}^+$ is the *teaching coefficient*, a coefficient weighting the losses against each other.

### 3.4 TRAINING THE TEACHER USING META-GRADIENTS

We train the teacher's weights $\phi$ using the the technique of *meta-gradient optimization*. This is done as follows: At the beginning of training, we split the dataset into a training and a validation set[4]. This split is kept during the entire duration of training. The split ratio is a hyperparameter.

The student's weights $\theta$ are updated $n$ times using Stochastic Gradient Descent on randomly sampled training batches, resulting in updated weights $\theta_n$. The student $\mathcal{S}$ is then evaluated on a validation set, producing a validation loss $L_{val}$. This validation loss is *optimized by the teacher*. This validation loss does not contain a term for the internal activation loss, but consists only of the label-loss $\mathcal{L}\left(\mathcal{S}(x_{val}), y_{val}\right)$. The teacher is optimized via the meta-gradient

$$\frac{\mathrm{d}L_{val}}{\mathrm{d}\phi} = \frac{\partial \mathcal{L}}{\partial \hat{y}}\left(\mathcal{S}(x_{val}; \theta_n), y_{val}\right) \frac{\partial \mathcal{S}}{\partial \theta}(x_{val}; \theta_n) \frac{\mathrm{d}\theta_n}{\mathrm{d}\phi} \tag{1}$$

where $x_{val}$ and $y_{val}$ are the inputs and targets from the validation set.

We omit the summation over individual loss components to avoid cluttering the notation. The crucial quantity $\frac{\mathrm{d}\theta_n}{\mathrm{d}\phi}$ describes how the final student's weights $\theta_n$ depend on the teachers weights $\phi$. It can be computed in linear computation time and space in the number of steps in the inner optimization loop using automatic differentiation[5]. The meta-gradient is then used for one step of stochastic gradient descent of the teacher's weights $\phi$. The student's weights are reset to what they were at the beginning of the step, since $\theta_n$ was only a hypothetical parameterization used to determine the meta-gradient. Then, the student is actually trained using the newly updated teacher for a certain number of steps. In our experiments, every step of meta-training is followed by $N$ steps of training the student's weights where $N$ is a hyperparameter. Alg. 1 describes the teacher update formally.

---

**Algorithm 1** Teacher update

---

**Require:** $\theta$ : Student parameters
**Require:** $\phi$ : Teacher parameters
**Require:** $\eta$ : Inner-loop learning rate
**Require:** $\alpha$ : Teaching coefficient
**Require:** $n$ : Number of inner-loop steps
1: $\theta_0 \leftarrow \theta$       $\triangleright \theta$ is unchanged in teacher update
2: **for** $i$ in $\{1..n\}$ **do**
3:      $x, x^*, y \leftarrow$ next training batch
4:      $\hat{y}, h \leftarrow \mathcal{S}(x; \theta_i)$
5:      $h^* \leftarrow \mathcal{T}(x^*; \phi)$
6:      $L_h \leftarrow \mathcal{L}_h(h, h^*)$
7:      $L_y \leftarrow \mathcal{L}_y(\hat{y}, y)$
8:      $L_{\text{train}} \leftarrow L_y + \alpha L_h$
9:      $\theta_i \leftarrow \text{SGD}(\theta_{i-1}, L_{\text{train}}, \eta)$
10: $x_{\text{val}}, y_{\text{val}} \leftarrow$ validation data
11: $L_{\text{val}} \leftarrow \mathcal{L}_y(\mathcal{S}((x_{\text{val}}; \theta_n), y_{\text{val}})$
12: Update $\phi$ to reduce $L_{\text{val}}$      $\triangleright$ Eq. 1

---

## 4 EXPERIMENTS

We implemented LDT in PyTorch (Paszke et al., 2019) using *higher* by Grefenstette et al. (2019). Note that in all experiments, we distinguish between a *validation set* and a *test set*. The validation set is used to train the teacher's parameters. Therefore, in order to allow for fair comparison with non-meta-learning baselines, we train baselines with the full training set and for LDT we split this set into a training and a validation portion. The test set is separate from the validation set and is used only passively to track generalization metrics.

---

[4]Note that the validation set is separate from the *test set*, which is an independently sampled dataset used only to evaluate the generalization performance of all methods.

[5]See Baydin et al. (2018) for a survey of Automatic Differentiation in Machine Learning.

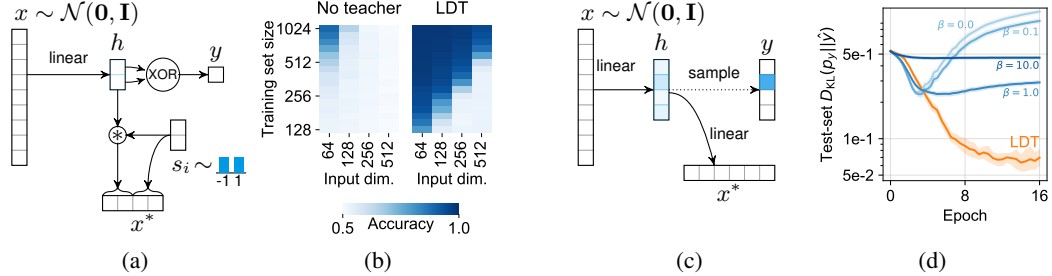

Figure 3: (a) Data generation diagram of task A. (b) Test accuracies on task A. For every combination, we report the maximum test accuracy achieved, averaged over 5 random seeds. (c) Data generation diagram of task B. (d) Test losses on task B achieved by LDT and the no-teacher baselines with different entropy regularization coefficients $\beta$.

## 4.1 Toy examples

Before moving on to datasets of dynamical systems, we study two toy tasks in order to give a better intuition for situations where privileged data can improve learning even though it is unavailable at test time, and at the same time to examine whether LDT can make use of the privileged data.

**Task A**    This task demonstrates a situation where the privileged information $x^*$ predicts the label $y$ perfectly and is lower-dimensional than the input $x$. At the same time, $x^*$ is not deterministically predictable from $x$. Formally, we construct the distribution over $x$, $x^*$, and $y$ such that the conditional expectation $\mathbb{E}[x^*|x] = 0$ for all $x$, and the conditional entropies $H(y|x) = H(y|x^*) = 0$. This is intended to correspond to a real-world setting where we observe training-time privileged data which gives us a low-dimensional explanation of the label, but this explanation has been obfuscated by noise and is hence unnecessarily difficult to predict directly (here impossible with a deterministic model).

We first sample a $D$-dimensional input $x$ with independent Gaussian components. This vector gets mapped to a 2-dimensional vector $h$ using a random but fixed linear transformation $A \in \mathbb{R}^{2 \times D}$. The label is obtained by applying XOR to $h > 0$. The privileged vector $x^*$ is constructed by independently sampling another two-dimensional vector $s$ which is multiplied with $h$ and concatenated to it. See Figure 3 for a diagram and Appendix B.1 for a detailed description of the dataset.

Using an MLP to predict $x^*$ from $x$ fails because the optimal predictor of $x^*$ from $x$ always outputs 0. However, in principle LDT can help in this situation: the teacher could learn to invert the stochastic mapping from $h$ to $x^*$. We set up a study to examine whether LDT automatically discovers a suitable inversion in practice. As student- and teacher-models we use MLPs with one hidden layer each. The teacher gets $x^*$ as input and produces target activations for the student's hidden layer. To investigate the sample efficiency, we let both the unguided student and LDT learn for a grid of different input dimensionalities $D$ and dataset sizes. The achieved test-set accuracies are shown in Figure 3, indicating that using a privileged data and a teacher makes this learning problem substantially more sample-efficient.

**Task B**    In this task (Figure 3c), instead of using deterministic labels and stochastic privileged data, we construct *deterministic* privileged data and *noisy* labels. A large neural network trained on these labels will tend to fit them exactly. As we discover empirically, this leads to ill-calibrated out-of-sample predictions.

An example for a practical situation where this applies is learning a value function in reinforcement learning using Monte Carlo or $n$-step temporal difference learning: environments and policies are typically stochastic, resulting in noisy empirical value targets. However, the trajectory of observations and actions as privileged data, can in principle explain away part of the influence of chance in the observed value target. We model this situation by providing as privileged data a transformed view of the logits that were used to sample the target. This transformation is unknown to the learner, and we investigate whether LDT can still use $x^\star$ to make the student learn a well-calibrated mapping.

We again sample the $D$ components of each input $x$ i.i.d. from $\mathcal{N}(0, 1)$. The random linear transformation $A \in \mathbb{R}^{d_h \times D}$ now transforms $x$ into a $d_h$-dimensional space. The privileged data $x^* \in \mathbb{R}^{d_p}$ is obtained via another linear transformation $B \in \mathbb{R}^{d_p \times d_h}$ of $h$. In our experiment we set $D = 128$, $d_h = 4$ and $d_p = 32$, and use 1000 training examples. The teacher gets $x^*$ as input and only needs to supervise the student's output layer. As baseline we train the student without a teacher or privileged data, but regularize its output predictions by subtracting $\beta H(\hat{y})$ from the training loss of each example, where $H(\hat{y})$ is the entropy of the model's prediction, and $\beta$ is a scalar coefficient. As shown in Figure 3d, learning the mapping from stochastic labels alone never learns a well-calibrated map from $x$ to $y$, while LDT learns to interpret the privileged information at training time, leading to a student that generalizes well at test time.

## 4.2 GAME OF LIFE

We performed an additional experiment aimed at evaluating whether LDT can help to extract dynamical information from trajectories. For that reason we created a dataset based on the cellular automaton Game of Life by John Conway. The input is a random initial state $x$, the output is the state of one particular cell after $n$ evolution steps. The privileged data $x^*$ consists of the trajectory of $n$ states after the first one. To make the task more difficult, $x^*$ is temporally permuted randomly, but consistently across examples. A detailed description is provided in Appendix C. As shown in Fig. 4, LDT can learn from the scrambled trajectory and help the student learn with less data than a model-free method.

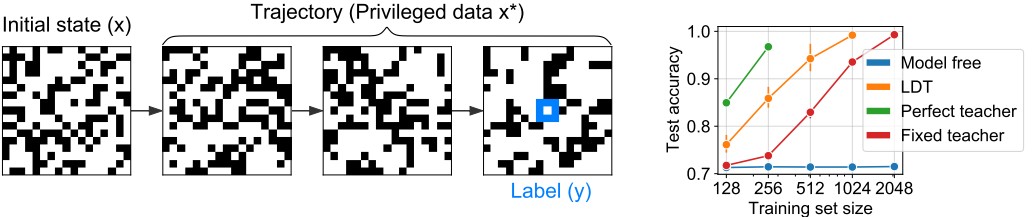

Figure 4: A datapoint in the Game-of-Life task (left) and the results of the experiment (right).

## 4.3 MuJoCo

In order to evaluate whether LDT can improve learning efficiency in continuous control tasks, we set up a prediction task using the MuJoCo simulator (Todorov et al., 2012). As an objective we choose learning an $n$-step reward model: such a model has to predict the sum of rewards along a trajectory of $n$ steps, given only access to the current state and $n$ (open-loop) actions. We fix $n = 16$ in all experiments.

We collect a dataset of size 4000 for each of the environments Swimmer-v2, Walker2d-v2, Hopper-v2, and HalfCheetah-v2. For each training example, the input $x = (s_t, a_{t:t+n})$ contains both an initial state and an action sequence. The initial state $s_t$ is obtained by executing a random policy for a short number of time steps after resetting the simulator. The action sequence $a_{t:t+n}$ consists of random actions. The label $y$ is the cumulative return of executing actions $a_{t:t+n}$ in state $s_t$. See Appendix B.2 for more details on the dataset.

**Models** We evaluate the following methods:

- **Model-free (MF)**: This baseline resembles the architecture shown in Fig. 1a. We use a five-layer fully connected MLP with ReLU activations and 128 neurons per layer. This baseline is trained with $x$ as input and $y$ as output.

- **Auxiliary task (Aux)**: As shown in Fig. 1c, this baseline augments the model-free method with an auxiliary task-head, which is trained to fit the full trajectory $x^*$ in order to shape the model's internal representation.

- **LDT**: Using the same MLP architecture as in the model-free baseline for the student, LDT additionally uses a teacher, which uses a network with a 1D-convolutional torso and an MLP-head to embed the trajectory $x^*$ and provide training signals for the student activations. The teaching

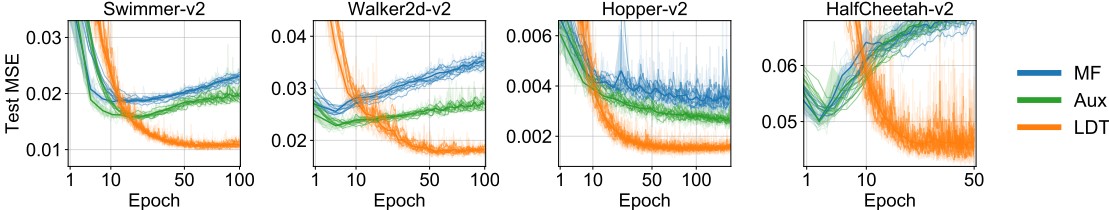

Figure 5: Test losses for the MuJoCo reward prediction task. Evolution of mean squared error between predicted and true normalized $n$-step reward on a held-out test set. We ran each configuration with 8 different random seeds and show the aggregated curves.

loss is a gated mean squared error $L_h \propto \sum_k \sigma(m_k)(h_k - h_k^*)^2$, where $\sigma$ is the logistic sigmoid function, $m$ and $h^*$ are the outputs by the teacher, and $h$ are the student's internal pre-activations.

For all networks we follow Schrittwieser et al. (2019) in how we turn the regression task of predicting rewards into a classification task by binning the reward space (see Appendix for details). Hyperparameters for each method are optimized independently (see Appendix for ranges) for each method and task. We select the configuration with the lowest mean-squared-error on test data and re-run it 8 times with different random seeds.

**Results**   In Figure 5 we report the mean squared error between predicted and true cumulative reward. The student trained with LDT achieved lower MSE than both the MF and Aux baselines in all tasks. Moreover, we found the *generalization gap* to be significantly smaller for LDT (Figure 6). This can be explained as follows: If the student is overfitting to the training set, its performance will degrade on the validation set. Since this loss only affects the teacher, the teacher can provide teaching targets $h^*$ that steer the student away when it starts to severely overfit to the training data.

We acknowledge that there could be strong baselines from the literature that we have not considered. Many of these approaches have complex pipelines of operations (e.g. Chua et al. (2018)), or use domain-specific knowledge to extract good representations (e.g. CPC (Oord et al., 2018)). As the main point of this paper is to investigate and understand a new framework to incorporate trajectory information, we decided to keep our evaluation setting simple and consistent, e.g. by using the same student architecture for *all* models.

## 4.4   ABLATIONS

We perform several ablation studies, in order to test the role of every component in our set-up. We describe every experiment set-up and show the results for the environment Walker2d in Figure 7. Results for the other environments are consistent with these, and are shown in Appendix B.2.4.

In the following list, we describe the different ablation studies in detail.

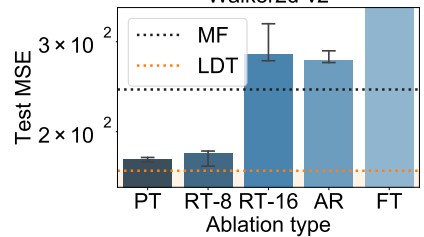

Figure 7: Ablation study results

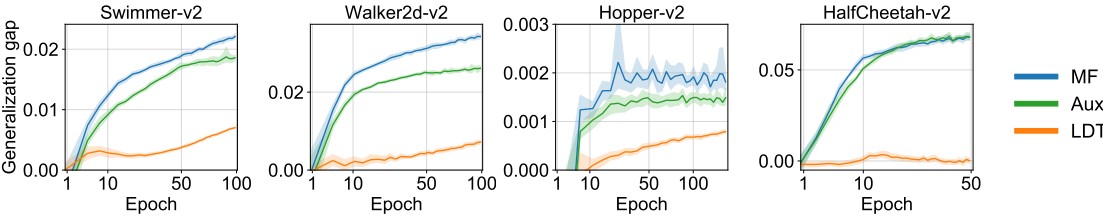

Figure 6: Generalization gaps (Test-set MSE minus Training-set MSE) for the different approaches and domains.)

(i) *Fixed untrained teacher (FT):* The teacher still provides target activations for the student, but we keep the teacher's weights fixed to the initial random weights. Using no meta-gradient updates, the student essentially fails to learn, validating the need to train the teacher.

(ii) *Time-permuted $x^*$ (PT)*: we independently and randomly permute the time order of $x^*$ for each example. Interestingly, the performance only degrades slightly. Two possible interpretations of this result are the following: either the teacher architecture does not make sufficient use of the temporal structure, or this structure is not so relevant for the task. Since we did not heavily tune our teacher network, we tend to lead towards the first explanation.

(iii) *Random $x^*$ (RT-$n$)*: each entry of the last $n$ frames of all trajectories $x^*$ are replaced by noise sampled i.i.d. from $\mathcal{N}(0, 1)$. When the trajectories are completely irrelevant to the task, the teacher can in principle learn to ignore them. However, since LDT trains with a smaller effective training set (because a portion is split off for validation), we expect a slightly weaker performance. Indeed, the results show that LDT with fully randomized trajectories (RT-16) tends to be comparable to, but slightly worse than the model-free network.

(iv) *Same training and validation data (AR):* instead of fixing the split of the training data into disjoint subsets for the student and for the teacher, we resample the split after every step of meta-training. Results show that the performance is even slightly worse than the model-free baseline. This is consistent with expectations, as the teacher's loss can be minimized directly by the student in training.

## 5 CONCLUSION

In this paper, instead of proposing a new hand-designed strategy to incorporate information from previous trajectories, we proposed to let a model *learn if and how* to use them. A teacher network learns to extract relevant information from a trajectory, and distills targets activations for a student network that only sees the current observation. The teacher is rewarded for maximizing the student performance on validation data, but can only achieve this indirectly by supervising the student while it trains on the training data. The aim of this method is to preserve advantages from both model-based as well as model-free methods: using the rich amount of information from observations as a training signal instead of just reward signal, but is not capped in performance due to model bias. One advantage that is *not* preserved from model-based approaches is the straightforward possibility to change tasks by adapting the reward function only. As validated empirically by the experiments and ablations presented in Section 4, this framework allows the model to choose where to sit in the wide spectrum between model-based and model-free methods, adapting to the specifics of the task at hand. An obvious drawback of LDT is — like many algorithms that learn how to learn — that computing meta-gradients increases the time and space complexity at training time by a factor that is linear in the number of inner steps. However, the computational cost of the student at test time is exactly the same as for an equivalent student that did *not* make use of the privileged information at training time, as the teacher does not play any role and can be discarded.

We believe that the general framework of teacher-student trained with meta-gradients to incorporate privileged information can be a fruitful direction for future work, beyond learning from trajectories. As the main limitation is the linear increase in time and space complexity at training time, increases in computing power should allow for more and more complex teachers and students to be trained on large tasks.

## ACKNOWLEDGMENTS

We wish to thank Lars Buesing and Arthur Guez for the fruitful discussions on the topic of this project. We thank the International Max Planck Research School for Intelligent Systems for supporting Alexander Neitz, and the Max Planck ETH Center for Learning Systems for supporting Giambattista Parascandolo.

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

## A    Implementation details

We implemented Meta-LUPI using PyTorch (Paszke et al., 2019) and used the library *higher* (Grefenstette et al., 2019) to compute meta-gradients.

Our implementation of LDT contains a number of hyperparameters and settings which we describe here.

- **Inner-loop optimizer**: The optimizer used on the student's weights $\theta_k$ in the inner loop of determining the meta-gradient (includes **inner learning rate** and **inner momentum**)
- **Student optimizer**: The optimizer used to actually update the student's weights $\theta$. (Includes **student learning rate** and **student momentum**)
- **Meta-optimizer**: The optimizer used to optimize the teacher's weights $\phi$. (Includes **meta learning rate** and **meta momentum**)
- **Validation split**: The fraction of the training set that is used to compute the meta-loss.
- **Teacher architecture**: The architecture of the teacher network.

## B    Experiment details

### B.1    Toy datasets

**Task A**    Construction of a training example:

- Sample $x \in \mathbb{R}^D$. $x \sim \mathcal{N}(\mathbf{0}, \mathbf{I})$
- Sample $A \in \mathbb{R}^{2 \times D}$ with $A_{ij} \overset{i.i.d.}{\sim} \text{Uniform}(\frac{1}{\sqrt{D}}, \frac{1}{\sqrt{D}})$ (default for `torch.nn.Linear`)
- Sample $s \in \{-1, 1\}^2$. $s \overset{i.i.d.}{\sim} 2 \cdot \text{Bernoulli}(0.5) - 1$
- $h = Ax$
- $y = \mathbf{I}[h_1 > 0] \oplus \mathbf{I}[h_2 > 0]$ where $\oplus$ denotes logical XOR.
- $x^* = [s_1, \ s_2, \ s_1 h_1, \ s_2 h_2]$
- Observed at training time: $(x, x^*, y)$
- Observed at test time: only $x$.

We perform an experiment over a grid of values for $D$ and $n$, where $n$ is the number of training examples in the dataset. The range of values for $D$ is the set $\{64, 128, 256, 512\}$. The range of values for $n$ are 16 exponentially-spaced integers between 128 and 1024. The student is a multi-layer-perceptron (MLP) with 32 and 128 neurons in the hidden layers, respectively. The architecture mirrors the data-generating process with a bottleneck after the first linear transformation - in particular, there is no activation function after the first hidden layer. We set up a baseline without a teacher, letting the student learn to predict $y$ from $x$. For the teacher $\mathcal{T}$ in LDT, we use an MLP with $[256, 256]$ neurons in the hidden layers, ReLU activations, and an output size of 32, matching the size of the student's first hidden layer. The *teaching loss* is the mean squared error between the teacher's output and the student's first hidden activation. Additionally, the student is always trained to minimize binary cross-entropy between its prediction and the training label. We use Adam with a learning rate of $10^{-3}$ as the optimizer for the student in both methods. For LDT, we use a validation split of $0.5$.

**Task B**    Construction of a training example:

- Sample $x \in \mathbb{R}^D$. $x \sim \mathcal{N}(\mathbf{0}, \mathbf{I})$
- Sample $A \in \mathbb{R}^{d_h \times D}$ with $A_{ij} \overset{i.i.d.}{\sim} \text{Uniform}(-\frac{1}{\sqrt{D}}, \frac{1}{\sqrt{D}})$
- Sample $B \in \mathbb{R}^{d_p \times d_h}$ with $B_{ij} \overset{i.i.d.}{\sim} \text{Uniform}(-\frac{1}{\sqrt{d_h}}, \frac{1}{\sqrt{d_h}})$
- $h = Ax$

- Sample $y \in \{0..d_h - 1\}$: $y \sim \mathrm{Categorical}(\mathrm{Softmax}(h))$
- $x^* = Bh$
- Observed at training time: $(x, x^*, y)$
- Observed at test time: only $x$.

For this experiment, we set $D = 128$ $d_h = 4$, $d_p = 32$, and use 1000 training examples.

The student is an MLP with input dimension $D$, two layers of 256 hidden units each, ReLU activations, and output dimension $d_h$. The teacher is an MLP with input dimension $d_p$, two layers of 256 hidden units each, ReLU activations, and output dimension $d_h$. As teaching loss, we use the KL-divergence between the softmax of the teacher's output and the softmax of the student's output activations.

Other hyperparameters used in this experiment are summarized in Table 1.

Table 1: Hyperparameters for LDT in toy task B

| Hyperparameter | Value |
|---|---:|
| Inner optimizer | Adam |
| Inner learning rate | 1e−3 |
| Batch size | 32 |
| Meta-optimizer | Adam |
| Meta-learning rate | 1e−3 |
| Teaching coefficient $\alpha$ | $10^4$ |
| Inner loop optimization steps $n$ | 64 |
| Validation split | 0.5 |

## B.2 MuJoCo

We now describe additional details for the MuJoCo reward-prediction experiments.

### B.2.1 Dataset details

The datasets for the MuJoCo reward prediction task are generated as follows.

We use the MuJoCo environments implemented in OpenAI gym Brockman et al. (2016), Each training set consists of 4000 examples generated as follows.

- Reset the environment to an initial state drawn from the initial-state-distribution
- Sample an integer $n$ uniformly from the range $\{10..30\}$
- Perform $n$ steps following random policy $\pi$.
- Record the current state $s$.
- Sample a 16 step open-loop action sequence $a_{1:16}$ from $\pi$.
- Set the example's input $x = (s, a_{1:16})$
- Execute $a_{1:16}$ starting from the current state $s$
- Record the trajectory of states $s_{1:16}$ and rewards $r_{1:16}$ as privileged data $x^*$
- Record the sum of rewards as label $y = \sum_t r_t$

As random policy $\pi$ we choose the policy that ignores the state and at each step and for each action dimension, samples one of the numbers $\{-1, 0, 1\}$ with equal probability, independently of each other.

We did not investigate the effects of using a different policy $\pi$ to generate the dataset.

The test sets consist of 10k examples distributed identically to their respective training sets.

Similarly to Schrittwieser et al. (2019) we turn the regression task of predicting rewards into a classification task by binning the reward space. We first obtain a transformation $\psi : \mathbb{R} \to [0, 1]$ in

such a way that it transforms training rewards to a uniform distribution in $[0, 1]$, interpolating linearly in-between values from the training set and clipping values outside the training-reward-range to lie between 0 and 1. We apply this transformation to both training and test labels and afterwards distribute the resulting values into 32 equally spaced bins between 0 and 1 to obtain categorical values. This leads to an even label distribution on the training set, and a roughly-even distribution on the test set. The label-loss-function is then the cross-entropy between predicted label probabilities and the one-hot distribution of the true label. The normalized mean-squared-error reported in the curves is obtained by first obtaining the expected value of the prediction by weighting the output-bucket with the predicted probability and then determining the squared difference to the true bucket value between 0 and 1.

Before training, we standardize all inputs along the dimensions individually, using empirical means and standard deviations found in the training set. Each state along the trajectory is standardized with the same normalization parameters.

### B.2.2 CHOOSING HYPERPARAMETERS

In order to determine good general ranges of hyperparameters[6], we first performed manual hyperparameter investigations and used a Tree-structured Parzen Estimator (Bergstra et al., 2011) search to find good ranges for the hyperparameters. Hyperparameter searches are performed using different random seeds from the final evaluation in order to reduce overfitting due to the hyperparameter-optimization. The curves for LDT shown in Figure 5 are the result of re-running the best hyperparameter configurations eight times with different random seeds. The curves for the baselines are obtained by performing a grid search on the hyperparameters shown in Tables 2 and 3, selecting the configuration that yielded the lowest test-MSE at any point in training and re-running it eight times with different random seeds.

We fix as architecture for the prediction model (the student in the LDT framework) a five-layer fully connected MLP with ReLU activations and 128 neurons per layer.

**LDT parameters** Tables 5 and 6 show hyperparameters we determined to work well for the MuJoCo reward prediction task.

As optimizers for the teacher and student weights, we use Adam (Kingma and Ba, 2015). For the inner-loop student-optimizer we use SGD with momentum.

As the teacher network we use a 1D-convolutional Neural Network followed by a fully connected network as follows: The 16 states of the trajectory $x^*$ (including rewards at every step) are fed into $n_{\text{conv}}$ 1D-convolutional layers, each followed by a ReLU activation. The first $n_{\text{conv}} - 1$ convolutional layers have $c_1$ output-channels, the last one has $c_2$ output-channels. The output of the last convolutional layer is flattened and fed into a fully connected downstream model with one layer of $c_3$ neurons.

In our experiments we found that it did not help to feed the actions along the trajectory to the teacher (additionally to the student who always gets to see the actions).

The output of the teacher is a vector that contains two entries for every neuron in the student network. One of these values, $h_k^*$, is the target-pre-activation, and $m_k$ is a gating parameter to weight that particular neuron's loss. We found it helpful to scale and shift the input to the gating-sigmoid by two constant scalars that are used for the entire network: the loss for a given pre-activation is computed as $\sigma(\frac{m_k}{2} - 1)(h_k - h_k^*)^2$, where $\sigma$ is the logistic sigmoid function, $h_k$ is the pre-activation in the student's network, $h_k^*$ is the target activation given by the teacher, and $m_k$ is the gating signal output by the teacher.

We use batch-normalization (Ioffe and Szegedy, 2015) in the teacher network and weight-normalization (Salimans and Kingma, 2016) in the student network, following the findings of Such et al. (2019).

As a precaution against exploding meta-gradients, we clip each component of the meta-gradient to the range $[-1e8, 1e8]$, but did not investigate whether this was necessary.

---

[6]By *hyperparameters* we mean those parameters which are fixed over the course of a training run. They do not consist of the teacher's *meta-parameters*.

Table 2: Hyperparameter ranges for the method 'model-free' (MF)

| Hyperparameter | Range |
|---|---|
| Learning rate | $\{1e{-}3, 1e{-}2\}$ |
| Weight decay | $\{0, 1e{-}5, 1e{-}4\}$ |
| Batch size | $\{8, 16, 32\}$ |

Table 3: Hyperparameter ranges for the method 'auxiliary' (AUX)

| Hyperparameter | Range |
|---|---|
| Learning rate | $\{1e{-}3, 1e{-}2\}$ |
| Weight decay | $\{0, 1e{-}5, 1e{-}4\}$ |
| Auxiliary task weight | $\{1, 2, 4\}$ |

Table 4: Hyperparameter ranges for our proposed method (LDT)

| Hyperparameter | Range |
|---|---|
| $\beta_1$ of Meta-optimizer (Adam) | $\{0.0, 0.9\}$ |
| Learning rate of Inner-loop optimizer (SGD) | $\{1e{-}3, 5e{-}3, 1e{-}2\}$ |
| Teaching coefficient ($\log_{10}$) | $\{2.0, 2.5, 3.0\}$ |
| Validation split | $\{0.3, 0.5, 0.7\}$ |

Table 5: Best hyperparameters found in grid search for LDT

| Hyperparameter | Swimmer-v2 | Walker2d-v2 | Hopper-v2 | HalfCheetah-v2 |
|---|---|---|---|---|
| $\beta_1$ of Meta-optimizer (Adam) | 0.9 | 0.9 | 0.0 | 0.0 |
| Learning rate of Inner-loop optimizer (SGD) | 1e$-$2 | 1e$-$2 | 1e$-$2 | 1e$-$2 |
| Teaching coefficient ($\log_{10}$) | 2.5 | 3.0 | 2.5 | 2.5 |
| Validation split | 0.3 | 0.5 | 0.3 | 0.7 |

Table 6: Other hyperparameters for LDT (all environments)

| Hyperparameter | Value (all envs) |
|---|---|
| Learning rate of Meta-optimizer (Adam) | 5e$-$4 |
| Learning rate of Student-optimizer (Adam) | 1e$-$3 |
| Batch size | 24 |
| Momentum of inner-loop optimizer (SGD) | 0.75 |
| Weight decay (L2) coefficient in inner-loop | 1e$-$8 |
| Number of inner-loop steps ($n_{\mathrm{inner}}$) | 96 |
| Teacher $c_1$ | 96 |
| Teacher $c_2$ | 256 |
| Teacher $c_3$ | 768 |
| Teacher $n_{\mathrm{conv}}$ | 4 |
| Number of student training steps per meta-step | 24 |

**Performances** When using the hyperparameters described above, we obtain the minimum test MSEs in the eight evaluation runs shown in Table 7.

### B.2.3 MODEL-BASED-BASELINES

A naive implementation of model-based could not fit the data. We trained the model within the same teacher-student framework, with a fixed dummy teacher that supervises the output of the student with the next observation (essentially mimicking a model-based setting). We 'stacked' the student such that it takes as input its output observation on the previous timestep, and by feeding it only

Table 7: Minimum mean squared errors on all environments

|                | no-teacher | auxiliary | LDT         |
|----------------|------------|-----------|-------------|
| HalfCheetah-v2 | 0.0485     | 0.0477    | **0.0427**  |
| Hopper-v2      | 0.00270    | 0.00224   | **0.00132** |
| Swimmer-v2     | 0.0169     | 0.0146    | **0.00964** |
| Walker2d-v2    | 0.0238     | 0.0211    | **0.0164**  |

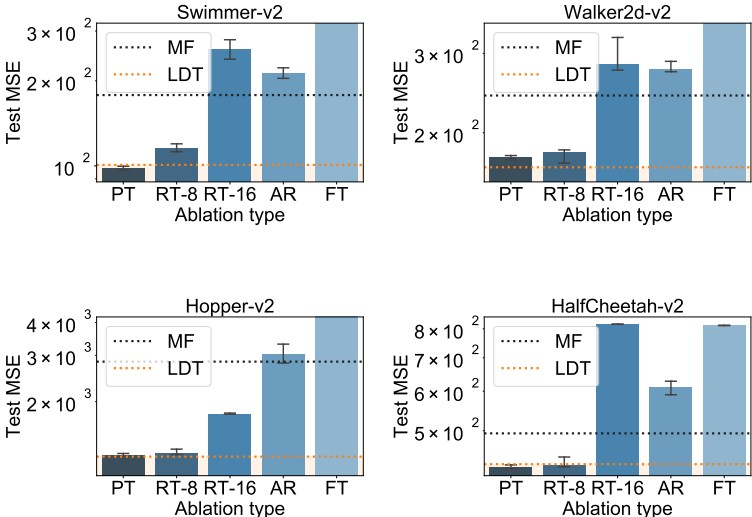

Figure 8: All results of ablation studies.

the first observation of the sequence, we supervise all of its intermediate targets. However, without introducing the extra inductive bias of training the student on every step *independently* of the others (instead of a single trajectory), the student's output would tend to diverge as it got deeper into the number of steps.

Training the student on pairs of consecutive transitions independently of the whole trajectory, makes the model work much better. However, making a fair comparison to the model-free, Aux, and LDT students is difficult, since the model-based student effectively uses $n = 16$ times more computation. Note that the main objective of this paper is to compare to what extent the *abstract* models implicitly learned by the *same* student architectures but with different techniques, can learn to incorporate the trajectory information. On this basis, we exclude the model-based baselines from our comparisons.

### B.2.4 ABLATION STUDIES

In Figure 8 we present the results of the ablation studies described in Section 4.4.

## C GAME OF LIFE EXPERIMENT

Here we provide details about the Game of Life experiment mentioned in Section 4.2.

**Dataset** As the underlying dynamical system, we use the cellular automaton *Game of Life* by John Conway. The rules of this cellular automaton are:[7]

1. Any live cell with fewer than two live neighbours dies, as if by underpopulation.

---

[7]Copied from `https://en.wikipedia.org/wiki/Conway%27s_Game_of_Life`.

2. Any live cell with two or three live neighbours lives on to the next generation.

3. Any live cell with more than three live neighbours dies, as if by overpopulation.

4. Any dead cell with exactly three live neighbours becomes a live cell, as if by reproduction.

Note that there is no linear decision boundary to determine the next cell state if the input space is the cell's neighborhood. However, with at least one hidden layer, it is possible to implement the rules. We can therefore use a convolutional neural network with $2n$ layers to represent the system's dynamics unrolled over $n$ steps.

We generate a binary classification task from the system as follows.

- A board of size $17 \times 17$ is initialized by setting each cell to "alive" with a probability of $p_I = 0.4$, independently of other cells. Alive cells are represented with a 1, all other cells are represented with a 0. This first board state is the input $x \in \{0, 1\}^{17 \times 17}$ for the task.

- The Game-of-Life rules are applied three times in sequence, yielding three consecutive states

- The classification target is the *middle cell in the last state*, i.e. a binary label

- All intermediate and final states in the trajectory are considered *privileged information*.

The initial alive-probability was chosen such that there typical rollouts are diverse. If the initial configuration is much sparser or much denser, then the population quickly dies off.

**Network architecture**  In early, unstructured experiments it seemed as though 3 layers per step make it easier to train a CNN using internal supervision. Therefore, we fix three convolutional layers per step for all experiments.

The student's internal activations $\mathbf{z}$ are convolutional feature maps. We use $\mathbf{z}_{kl}$ to denote the $l$'th feature map of the $k$'th layer. Each $\mathbf{z}_{kl}$ has the same dimensions as the map of the cellular automaton ($17 \times 17$ in our experiments).

**Implementation of the teacher**  We choose a simple parameterization of the teacher. The teacher's weights are a three-dimensional tensor $\phi \in \mathbb{R}^{T \times N \times F}$, where $T$ is the fixed number of steps in the Game-of-Life trajectory, $N$ is the number of convolutional hidden layers in the student network, and $F$ is the number of feature maps per hidden layer.

The internal activation targets are linear combinations of the cell states in the trajectory $(x_1, ..., x_T)$:

$$h_{kl}^* = \sum_t a_{tkl} x_t \tag{2}$$

where $a_{\bullet kl} = \text{softmax}(\phi_{\bullet kl})$, i.e. the mixture weights are the teacher's weights softmaxed-through-time. More explicitly,

$$a_{tkl} = \frac{\exp(\phi_{tkl})}{\sum_\tau \exp(\phi_{\tau kl})} \tag{3}$$

We use Adam to update the inner weights and vanilla SGD+momentum to update the meta-parameters, as we found these two choices to generally perform best.

**Baselines**

- **Fixed oracle teacher**: $\phi$ is initialized to the values such that it provides the frames in the correct sequence to every third convolutional layer. It is held fixed over the course of training.

- **Fixed random teacher**: $\phi$ is initialized randomly and held fixed over the course of training.

- **No teacher**: Classification task without privileged data

**Experimental protocol**   The questions that this experiment should help answer is: Does meta-gradient training help with sample efficiency in the Game-of-Life task compared to using (a) no teacher, (b) a fixed random teacher?

- Training set sizes: $\{128, 256, 512, 1024, 2048, 4096, 8192\}$, but in order to be resource-efficient, try the approaches from fewer examples to more examples until they "max out".
- Methods: meta-learned teacher (*meta*), fixed random teacher (*fixed-teacher*), perfect teacher (*perfect-teacher*), no teacher (*no-teacher*)
- In meta-learning, the train-validation-split happens within the training set (no additional validation data is provided).
- We use the same hyperparameter ranges for all experiments - ones that seem reasonable based on initial experimentation.
- Train for 200 epochs.

| Hyperparameter | Distribution | Min | Max | Methods |
|---|---|---|---|---|
| learning_rate | LogUniform | $10^{-4}$ | $10^{-2}$ | all |
| adam_momentum | 1.0 - LogUniform | $10^{-1}$ | $10^{0}$ | all |
| weight_decay | LogUniform | $10^{-8}$ | $10^{-3}$ | all |
| batch_size | DiscreteUniform | 32 | 128 | all |
| weight_init_multiplier | LogUniform | 0.1 | 2.0 | all |
| n_filters | DiscreteUniform | 16 | 32 | all |
| internal_coef | LogUniform | $10^{2}$ | $10^{4}$ | meta, fixed-teacher |
| teacher_weight_scale | LogUniform | $10^{-2}$ | $10^{0}$ | fixed-teacher |
| teacher_attention_scale | LogUniform | $10^{-3}$ | $10^{-1}$ | fixed-teacher |
| n_inner | DiscreteUniform | 16 | 32 | meta |
| meta_learning_rate | LogUniform | $10^{-6}$ | $10^{-3}$ | meta |
| meta_momentum | LogUniform | $10^{-2}$ | $10^{0}$ | meta |
| validation_split | LogUniform | $10^{-2}$ | $10^{0}$ | meta |

**Hyperparameter ranges**   The parameters for the initial teacher weights are used exclusively by the fixed-teacher method.

Note that while the meta-learning approach has more hyperparameters than the baselines, the overall computational budget given to each method is the same - more hyperparameters are not advantageous by default.

We run a hyperparameter search over the specified range for each combination of (method, n-training-examples). As hyperparameter optimization algorithm we use a Tree-structured Parzen Estimator (Bergstra et al., 2011) for each method.

