# OpenReview forum: "A teacher-student framework to distill future trajectories"
_ICLR.cc/2021/Conference — ICLR 2021 Poster_

### Official Review · AnonReviewer4 · 2020-10-27
**Recommendation to Accept**

**Rating:** 6
**Confidence:** 4

**Review:**

This paper proposes a teacher-student training scheme to incorporate the useful information of trajectory to improve the predictive performance of model-free methods. The teacher network tries to "guide" the student network at the training stage by presenting an interpretation of the trajectory. The guidance is implemented by adding to the loss function a regularization term that penalizes the "distance" between the teacher's output and the hidden states of the student. The proposed method was tested and compared to other model-free methods.

The study in this work is interesting and important in RL. It tries to tackle the weakness of model-free methods by introducing the dynamics while avoiding to build a full model like the model-based methods. Searching for an optimal tradeoff between the two would benefit the practical uses.

I'd vote for accepting the manuscript if the authors could address my concerns.

- As mentioned in the text, the student internal h and the supervision signal h* are not required to have the same dimensionality. Are they required to have the same number of layers or certain correspondence between layers? Why or Why not? Do the authors have general principles on the design of the teacher and the teaching loss?

- Is there any particular reason for using classification loss in all examples especially for a typically regression problem? Did regression loss lead to bad performance?

- In the toy XOR example, the particular design of x* seems to play the key role. It is not the typical trajectory of x. This hand-crafted data does not help with demonstrating the teacher learning the powerful interpretation itself.

- Though the authors have talked about why not to compare to a model-based method, I do not think it is convincing. As mentioned in the intro section, the model-based methods fail due to partial observability and etc, but I do not see the examples in this study have such issues. The computation of model-based methods depends on the complexity of internal model rather than the task. The argument should then be if the proposed method outperform model-based methods given the same complexity (e.g. the number of parameters) or same amount of data.

---

> ### Author Response · Authors · 2020-11-18
> **Response to Reviewer 4**
>
> Thank you for your positive review! We are happy to address the concerns you raised:
>
> **Dimensionality of h\* and h:**
> Thank you for pointing this out — upon re-reading the submitted manuscript, we noticed that this part needs a clarification: In Section 3 we intentionally keep the specific choice of the teaching-loss-function open as an implementation detail. In principle, one could use any function that compares some transformation of the student’s activations with targets provided by the teacher. The supervision could affect some subset of the layers, etc. Exploring the space of teaching-loss functions is an interesting direction for future work.
>
> We decided to focus on what seems as the most straightforward setup, which is currently described in Section 4: We start by letting the teacher output a target-activation vector for every layer of pre-activations in the student network, and to define the teaching loss as the average element-wise squared error between pre-activations and targets. However, this would mean that the teacher is forced to provide a target activation for every neuron. We found that it helps to relax this requirement by letting the teacher output both a target activation as well as a “masking weight” $m_k$ which scales this loss component. This allows the teacher to leave certain neurons unsupervised (in an example-dependent way).
>
>
> **Is there any particular reason for using classification loss in all examples especially for a typical regression problem? Did regression loss lead to bad performance?:** We decided to use a classification loss based on the results from MuZero (Mastering Atari, Go, Chess and Shogi by Planning with a Learned Model, Schrittwieser et al.), that observed more stable results using cross-entropy instead of mean squared error for rewards and values. Therefore, we did not try using the regression loss.
>
>
> **(1) As mentioned in the intro, model-based methods fail due to partial observability and etc, but I do not see the examples in this study have such issues. + (2) In the toy XOR example, the particular design of $x^\*$ seems to play the key role. It is not the typical trajectory of $x$. This hand-crafted data does not help with demonstrating the teacher learning the powerful interpretation itself.**
>  We agree that the MuJoCo and Game-of-Life tasks are fully observed and therefore less of a problem for model-based methods than partially observed tasks. On the other hand, the XOR example you mentioned was precisely chosen to demonstrate a simple case where a deterministic model fails to predict the privileged data.
>
> In Appendix B2.3 we detailed that making a fair comparison to the model-free, Aux, and LIT students is difficult, since the model-based student effectively uses n=16 times more computation. Note that the main objective of this paper is to compare to what extent the abstract models implicitly learned by the same student architectures but with different techniques, can learn to incorporate the trajectory information. On this basis, we did not include the model-based baselines in our comparisons.

---

### Official Review · AnonReviewer2 · 2020-10-28
**well motivated**

**Rating:** 6
**Confidence:** 2

**Review:**

Summary:
This work tries to find a compromise of model-based and model-free methods, using a teacher and student network . The teacher network is trained with meta-gradients. It interprets the trajectories and provides activations for a student network that is supervised for a given task using the current state.

Strengths:
+ The paper is well motivated and solves interesting problem.
+ The related work is thoroughly reviewed.

Weaknesses:
- Some claims made by authors are not validated. I suggest to add relevant citations in Sec.1. These claims support the motivation of this work but are not acceptable without proper references. For example, where is the evidence of deterioration on the tasks that are potentially relevant? Which task do model-free methods achieve substantially better performance?
- The evaluation is conducted using the self-generated baselines. Why don't you use existing methods shown in Sec.2 and compare? I cannot find the result of the model-based baseline (as a counterpart of MF).
- Typos: using only using model-free methods

---

> ### Author Response · Authors · 2020-11-18
> **Response to Reviewer 2**
>
> Thank you for the positive review, we are glad you found the work to be well-motivated and interesting.
>
> **I suggest to add relevant citations in Sec.1.**
> **(1) "deterioration on the tasks that are potentially relevant"?**
> We reworded the statement about the deterioration on relevant tasks as it was somewhat ambiguous.
> **(2) Which task do model-free methods achieve substantially better performance?**
> One of the main benefits of model-based methods is sample-efficiency. Given sufficient data (often several orders of magnitude more), current model-based methods are sometimes outperformed by model-free methods. Examples where model-free methods achieve state-of-the-art asymptotic performance include challenging tasks such as the game StarCraft (AlphaStar) and robotic manipulation tasks such as the described in the Rubik’s cube paper by OpenAI (we included these references in the introduction).
>
> **Why don't you use existing methods shown in Sec.2 and compare? Where is the MB baseline?** We should clarify this aspect in the manuscript, as AnonReviewer1 shared your first question. The baselines we chose are fairly simple and established methods. The reasons for our choice were:
> i) The approach we propose is fairly generic and orthogonal to most techniques used to regularize model-free training.
> ii) The main other method that tries to address a similar problem to ours is Value driven Hindsight Modeling (Guez et al., NeurIPS 2020): while a comparison to this would be interesting, the work is very recent, there is no publicly available source code released at the moment, and it would be hard to obtain fair and sound results when comparing to it.
> We focused on analysing the new method we presented empirically, we decided instead to design several ablation studies, as they provide more insight into how the method performs.
>
> Regarding the model-based baseline, in Appendix B2.3 we detail that making a fair comparison to the model-free, Aux, and LIT students is difficult, since the model-based student effectively uses n = 16 times more computation. Note that the main objective of this paper is to compare to what extent the abstract models implicitly learned by the same student architectures but with different techniques, can learn to incorporate the trajectory information. On this basis, we did not include the model-based baselines in our comparisons.
> The typo is now fixed, thanks!
> Do you still have any concern or feedback that we could incorporate?

---

### Official Review · AnonReviewer3 · 2020-10-28
**Interesting work, but the framing is confusing**

**Rating:** 6
**Confidence:** 3

**Review:**

This paper presents a student-teacher framework, where the teacher network can be used to select and prioritize the relevant properties of the given dynamical system that should be learned by the student.

Pros:
- (significance) I think the presented framework is a powerful one, and has a potential to be applied broadly to many real-world problems.
- (quality) The development of the method is sound and well-motivated. The method was tested on a toy example and then applied to tasks with varying degrees of challenges.

Cons: (mostly on clarity)
- What confuses me the most is the way the authors frames their work. From the current title "Learning to interpret trajectories", and the abstract, it was not really clear to me what the paper is about; I am not sure if the population of people who stop at the title would be the same as the population that finds the contents most interesting. Specifically:
    - Why is "trajectory" a central keyword for this work? The word trajectory can be used in many different contexts, I don't think it was made clear anywhere in the paper what is the defining features of the "trajectory-ness" that the authors want to emphasize.
    - What do you mean by "interpret"? This word is being used in a very loose fashion without a clear context; it is empty at best, and misleading at worst.
- I get that the proposed framework avoids the *problems* of model-based and model-free methods, but I am having difficulties identifying what *advantages* of the two methods that the framework is incorporating.
- One of the central questions that is raised in the introduction is this: "What is the right way to incorporate information from different trajectories?". But I am not sure how this work solves the problem of incorporating *different* trajectories specifically.

Additional comment:
- The core concepts from the previous works that the work is based on, such as *learning using privileged information* or the meta-gradient approach, are not clearly introduced. Even a brief, ~1 sentence description would be helpful.
- The ideas of model-based and model-free methods are reinforcement learning concepts, and may not be clear to people who are not in RL. Again some brief description would help.

Overall, I have a mixed feeling about this paper and I currently stand between scores 5 and 6. Whereas I find the proposed method interesting, I feel that the lack of clarity, and the confounding of messages in the framing, make this paper rather short of the standard for the conference.

**UPDATE:**
My major concerns were addressed in the revised version of the paper.

---

> ### Author Response · Authors · 2020-11-18
> **Response to Reviewer 3**
>
> Thank you for the extensive feedback. The main concerns you express are w.r.t. to clarity, and we agree that they need to be addressed.
>
> **Framing**: As you point out, the method itself is more general than exclusively requiring trajectories as privileged information. Upon deciding how to present the work, we were unsure whether it would be fair to claim more generality for it than applications to trajectories, given that this has been the main topic we have investigated in our experiments so far.
> As for the word "interpret", we meant it as "the teacher learns to find what's relevant in the privileged information, and distills/transforms/translates it into target activations for the student to fit".
> We understand your concern and we agree that a more suitable term should be found.
> Overall, we are happy to adjust the title and framing of the paper. How would you recommend presenting it?
> A few alternative titles that we have been considering (but we are happy to hear your proposals too):
> - "Learning to distill trajectories into implicit models"
> - "Teaching by distilling trajectories to learn implicit models"
> - "A teacher-student framework to distill trajectories into implicit models"
>
> We are aware that "distillation" is often used in the context of "distilling a large model into a smaller one", but combining it with “trajectories” might prevent the ambiguity.
>
> **The proposed framework avoids the problems of model-based and model-free methods, but what advantages of the two does it keep?:**
> From model-free it preserves the following advantages:
> - Its asymptotic performance is not capped by modelling errors of the environment, since there are no observations to precisely reconstruct, nor accumulating errors in longer rollouts.
> - It can learn to model the environment "internally" at any spatial or temporal resolution (e.g., it could internally learn to plan asynchronously)
>
> From model-based:
> - It can use the rich amount of information in future observations to bootstrap learning the task (instead of having to rely only on rewards/values). An example of this is the game-of-life dataset (Section 4.2), where model-free methods are too unconstrained and can find spurious explanations that are unrelated to the underlying mechanisms.
> In other words, LIT preserves the advantage of using more learning signal than model-free methods, which rely on a single scalar per example.
>
> One advantage that is not preserved from model-based is the straightforward possibility to change tasks by adapting the reward function only.
>
> Thanks for your comment, we made this more explicit in the conclusion section of the paper.
>
> **Even a brief, ~1 sentence description of learning from privileged information would be helpful.** We added this to the end of the introduction, thanks!
>
> **One of the central questions that is raised in the introduction is this: "What is the right way to incorporate information from different trajectories?". Why "different"?:**
> Thank you for pointing this out. By "different" trajectories, we meant incorporating information from "several" trajectories, not in the sense of simultaneous trajectories but in the sense of several examples each with one (future) trajectory. We had not seen the potential for a misunderstanding there. It is now fixed, thanks!
>
> We are glad you found the work to be significant and its quality high. We look forward to hearing your opinion about the title/framing, and we thank you again for helping us improve the clarity of the paper.

---

> > ### Comment · AnonReviewer3 · 2020-11-23
> > **Response to authors**
> >
> > Dear authors, thank you for the response; it clarified most of my questions.
> >
> > Regarding the clarity of the framing, thank you for being open to the feedback. To be very specific, I recommend that the authors:
> >
> > - make sure that all the words that they use in the title is appropriately put into context in the paper (preferably in the early part), and
> > - perhaps update the title itself to make it more informative about the work, as they are already considering.
> >
> > I do like the newly suggested titles better, because they are more specific about what the method does. For the same reason I think the version with "teacher-student framework" is most clear; but this may be my personal preference, so please do not feel obliged by this comment.
> >
> > What I would recommend more strongly, more than the specific wording of the title, is to make sure that no aspect of the title is left unclarified by the time a reader reads through the introduction. For example, the newly suggested titles all have the word "distill", but this word is currently not used in the paper text (not until the conclusion section), so it is context-less and vague. The way the authors illustrated the typical use of the term in their response ("distilling a large model into a smaller one") was great and very clear. The authors could either use the word "distill" in the paper like this so that its meaning in this context becomes clear, or speak in terms of other words that they actually use in the paper (e.g., there are several instances of "extract").

---

> > > ### Author Response · Authors · 2020-11-23
> > > **Thanks!**
> > >
> > > Dear reviewer, thank you for getting back to us. We integrated your recommendations as follows: We
> > > - updated the title to "A teacher-student framework to distill future trajectories";
> > > - removed all mentions of the word "interpret" and now only use "extract information from trajectories" and "distill into target activations";
> > > - clarified the use of the term "distill" in the first page;
> > > - updated every plot and mention of the algorithm, that we now call "LDT" for Learning to Distill Trajectories.
> > >
> > > Thanks again for helping us improve the clarity of the paper!

---

### Official Review · AnonReviewer1 · 2020-10-28
**A novel approach to predict labels of dynamical systems**

**Rating:** 6
**Confidence:** 3

**Review:**

This paper proposes a learning framework for predicting the labels of dynamic systems. Unlike existing model-based approaches and model-free approaches, the proposed model takes a middle ground and uses a knowledge distillation-based framework. It uses a teacher model to learn to interpret a trajectory of the dynamic system, and distills target activations for a student model to learn to predict the system label based only on the current observation.

Experimental results on both synthetic and simulated datasets confirm the effectiveness of the proposed framework.

Pros:
1. The paper studies an important problem. Predicting the behavior of a dynamic system has many applications.

2. The proposed model is interesting and may lead to a series of follow-up studies that leverage the strengths of both model-free and model-based methods using knowledge distillation techniques.

Cons:
1. The baseline models are quite simple. There are stronger baselines as noted by the authors. While the proposal is a learning framework, it might still be worth customizing and comparing it with state-of-the-art models in specific problems.

2. The presentation of the paper can be improved. It would be good to add a running example to explain the various concepts and definitions used in the paper.

Additional comments:
Typo: "a teacher networks" => "a teacher network"; "using only using" => "using only"

**Update after author response:** I appreciate the authors' efforts to address my comments. The new version reads better. However, I am still not entirely convinced by the choice of the simple baselines. Since a positive rating is already given, I would keep it unchanged.

---

> ### Author Response · Authors · 2020-11-18
> **Response to Reviewer 1**
>
>
> **Baselines:** We agree with you, the baselines we chose are fairly simple and established methods. The reasons for our choice were:
> i) The approach we propose is fairly generic and orthogonal to most techniques used to regularize model-free training.
> ii) The main other method that tries to address a similar problem to ours is Value driven Hindsight Modeling (Guez et al., NeurIPS 2020): while a comparison to this would be interesting, the work is very recent, there is no publicly available source code released at the moment, and it would be hard to obtain fair and sound results when comparing to it.
> We focused on analysing the new method we presented empirically, so we decided instead to design several ablation studies, as they provide more insight into how the method performs.
>
>
> **It would be good to add a running example to explain the various concepts and definitions used in the paper:** Thanks for pointing this out. Based on your comment, we added an example of medical decision-making to the manuscript (second and third paragraphs of Section 3.1) and use it to explain the definitions introduced in Section 3.1:
> The task is to predict whether a patent will recover under a certain treatment. The observations of the system could be measurements (biopsies, X-ray images, etc.) taken on the patient over time. In this task, we could choose to predict the outcome -- a binary variable -- based on the current and past measurements alone, but in doing so, we would throw away rich explanations that could give more clues about why the patient did or did not recover (pure model-free). On the other hand, modelling the future measurements conditionally on the past and present (pure model-based) might be a challenging task: for example, a medical report written in human language would be extremely hard to predict. The teacher network we propose in LIT can learn to extract the task-relevant information from all future measurements and convert them to a learning signal that guides the student toward better-generalizing solutions.
>
> The 2 typos you pointed out are now fixed, good catch!
>
> Thank you for your feedback, it helped us improve the presentation of the paper! We are glad you found our learning framework to be novel, that it studies an important problem with many applications and with ample room for follow-up studies that leverage the strengths of both model-free and model-based methods using knowledge distillation techniques.
>
> Are there any concerns left that we could address?

---

### Author Response · Authors · 2020-11-18
**Comment to all reviewers**

We thank all reviewers for their constructive feedback, which greatly helped improve the clarity of the paper.

We want to emphasize that beyond the conceptual framework that we introduced, the method we proposed could be widely applicable to a variety of problems in future work, including medical diagnosis and decision-making.

In an effort to improve reproducibility and to allow potential future work to build on ours, we commit to releasing the code with the final version of the paper.

---

### Decision · Program_Chairs · 2021-01-07
**Final Decision**

**Decision:**

Accept (Poster)

**Comment:**

The paper proposes a new teacher-student framework where the teacher network guides the student network in learning useful information from trajectories of a dynamical system. The proposed framework is inspired by the Knowledge Distillation method. The teacher learns what information should be used from the trajectories and distills this information for the student in the form of target activations. In a nutshell, the framework allows the student to interpolate between model-based and model-free approaches in an automated fashion. Experimental evaluation on both the hand-crafted and simulated tasks demonstrate the effectiveness of the proposed framework. The reviewers had borderline scores in their initial reviews and raised several questions for the authors. The reviewers appreciated the rebuttal, which helped in answering their key questions -- I want to thank the authors for engaging with the reviewers during the discussion phase. The reviewers have an overall positive assessment of the paper, and believe that the proposed teacher-student framework is novel and potentially useful for many real-world problems. The reviewers have provided detailed feedback in their reviews, and I would like to strongly encourage the authors to incorporate this feedback when preparing the final version of the paper.